# Low P66shc with High SerpinB3 Levels Favors Necroptosis and Better Survival in Hepatocellular Carcinoma

**DOI:** 10.3390/biology10050363

**Published:** 2021-04-23

**Authors:** Silvano Fasolato, Mariagrazia Ruvoletto, Giorgia Nardo, Andrea Rasola, Marco Sciacovelli, Giacomo Zanus, Cristian Turato, Santina Quarta, Liliana Terrin, Gian Paolo Fadini, Giulio Ceolotto, Maria Guido, Umberto Cillo, Stefano Indraccolo, Paolo Bernardi, Patrizia Pontisso

**Affiliations:** 1Department of Medicine, University of Padua, Via Giustiniani, 2, 35128 Padua, Italy; silvano.fasolato7@gmail.com (S.F.); mariagrazia.ruvoletto@unipd.it (M.R.); santina.quarta@unipd.it (S.Q.); liliana.terrin@gmail.com (L.T.); gianpaolo.fadini@unipd.it (G.P.F.); giulio.ceolotto@unipd.it (G.C.); mguido@unipd.it (M.G.); 2Istituto Oncologico Veneto IOV- IRCCS, 35128 Padua, Italy; giorgia.nardo@iov.veneto.it (G.N.); stefano.indraccolo@iov.veneto.it (S.I.); 3Department of Biomedical Sciences, University of Padua, 35131 Padua, Italy; andrea.rasola@unipd.it (A.R.); ms2122@mrc-cu.cam.ac.uk (M.S.); paolo.bernardi@unipd.it (P.B.); 4Department of Surgical, Oncological and Gastroenterological Sciences-DISCOG, University of Padua, 35128 Padua, Italy; giacomo.zanus@unipd.it (G.Z.); cillo@unipd.it (U.C.);; 5Hepatobiliary and Pancreatic Surgery Unit-Treviso Hospital, 31100 Treviso, Italy; 6Department of Molecular Medicine, University of Pavia, 27100 Pavia, Italy; cristian.turato@unipv.it; 7Unit of Hepatobiliary Surgery and Liver Transplantation, Padua University Hospital, 35128 Padua, Italy

**Keywords:** liver cancer, prognosis, animal experimental models, oxidative stress

## Abstract

**Simple Summary:**

Cell proliferation and escape from apoptosis are important pathological features of hepatocellular carcinoma, one of the tumors with the highest mortality rate worldwide. The aim of the study was to evaluate the expression of the pro-apoptotic p66shc and the anti-apoptotic SerpinB3 molecules in relation to clinical outcome in patients with hepatocellular carcinoma and to evaluate their effect on cell fate and tumor growth. In patients with hepatocellular carcinoma the best survival was observed in the subgroup with p66shc levels below median values and SerpinB3 levels above median values. Mice *p66shc* knockout showed high levels of SerpinB3, while in hepatoma cells overexpressing SerpinB3, p66shc expression was trivial. Hepatoma cells overexpressing SerpinB3 were more prone to die after oxidizing treatments. These cells injected in nude mice developed tumors five times smaller than those from controls. Tumors originating from hepatoma cells overexpressing SerpinB3 showed typical features of necroptosis. In conclusion, in patients affected by hepatocellular carcinoma, the pattern characterized by p66shc downregulation and elevated SerpinB3 levels was associated with markedly better survival. This pattern favored necroptosis in experimental high-stress conditions.

**Abstract:**

Cell proliferation and escape from apoptosis are important pathological features of hepatocellular carcinoma (HCC), one of the tumors with the highest mortality rate worldwide. The aim of the study was to evaluate the expression of the pro-apoptotic p66shc and the anti-apoptotic SerpinB3 in HCCs in relation to clinical outcome, cell fate and tumor growth. p66shc and SerpinB3 were evaluated in 67 HCC specimens and the results were correlated with overall survival. Proliferation and cell death markers were analyzed in hepatoma cells overexpressing SerpinB3, under different stress conditions. *p66shc^−/−^* mice and xenograft models were also used to assess the effects of p66shc and SerpinB3 on tumor growth. In patients with HCC, the best survival was observed in the subgroup with p66shc levels below median values and SerpinB3 levels above median values. Mice *p66shc^−/−^* showed high levels of SerpinB3, while in HepG2 cells overexpressing SerpinB3, p66shc expression was trivial. HepG2 overexpressing SerpinB3 cells were more prone to die after oxidizing treatments, such as diamide or high concentration H_2_O_2_. These cells injected in nude mice developed tumors five times smaller than those from control HepG2 cells. Tumors originating from HepG2 overexpressing SerpinB3 cells showed decreased activated Caspase-8, with concomitant increase of RIP3K and decreased levels of cleaved RIP3K, typical features of necroptosis. In conclusion, in patients affected by HCC, the pattern characterized by p66shc downregulation and elevated SerpinB3 levels was associated with markedly better survival. This pattern favored necroptosis in experimental high-stress conditions.

## 1. Introduction

Hepatocellular carcinoma (HCC) is the most common primary malignant tumor of the liver worldwide. Its pathogenesis is multifactorial, with a strong aetiological association with chronic viral infection by hepatotropic viruses, alcohol consumption, exposure to hepatic toxins, as well as genetic disorders such as haemochromatosis and α1-antitrypsin deficiency. Fine mechanisms eliciting tumor development are still poorly understood. In the promotion stage, HCC has been associated with defective apoptosis and increased cell proliferation [1,2]. Tumor cells often show altered expression of genes involved in cell proliferation and cell death [3]. Among them, the 66-kDa isoform of ShcA (p66shc) is an adapter protein that was proposed to downregulate mammalian life span by inhibition of receptor tyrosine kinase signaling and induction of cell differentiation [4,5]. This protein has been involved in cellular response to oxidative stress and apoptosis [5] and its pro-apoptotic functions have been widely confirmed, although its role is complex, since it could act as a double-edged sword in the regulation of apoptosis, depending on the environmental context [6,7,8].

Aberrant expression of p66shc could also be involved in various stages of carcinogenesis. Elevated levels of p66shc protein have been found in estrogen-regulated tumors, including metastatic breast and ovarian tumors, thyroid tumors and stage II colon cancer [9,10]. Recently, p66shc has been identified as a novel regulator of autophagy and apoptosis in B lymphocytes, allowing their survival and differentiation [11]. Very little information is available on p66Shc in HCC, even if some data have reported its correlation with a poor clinical outcome [12].

SerpinB3, formerly known as Squamous Cell Carcinoma Antigen-1 or SCCA-1, is a cysteine peptidase inhibitor, member of the ovalbumin family of serine proteinase inhibitors [13]. It is a multifunctional molecule, the activities of which have not yet been completely defined. Beside its anti-protease activity [14,15], this serpin has been reported to inhibit apoptosis through the interaction with mitochondrial respiratory complex I [16] and to increase proliferation, improving survival of tumor cells [17]. However, it has also been found that it can promote accelerated cell death through Caspase-8-mediated apoptosis in response to endoplasmic reticulum stress [18]. SerpinB3 has been found to be overexpressed in several types of tumor, especially in those with poor prognosis, including breast, liver, esophagus and colorectal cancer [19,20,21,22], although some findings have suggested its ability to inhibit cancer cell invasion [23]. These findings suggest that additional players might modulate the biological activity of this molecule, influencing cell fate.

To date, the available evidence is consistent with the existence of different forms of regulated cell death. Necrosis and apoptosis represent two pathways of genetically encoded necrotic cell death [24] and apoptotic cell death [25]. Necroptosis is a non-caspase-dependent and precisely regulated mechanism of cell death. Necroptosis serves as an alternative mode of programmed cell death overcoming apoptosis resistance and may trigger and amplify antitumor immunity in cancer therapy [26].

In this study we have analyzed the expression of p66shc and SerpinB3 in relation to overall survival in patients with primary liver cancer, and carried out in vitro and in vivo experiments to define the effect of these two molecules, involved in cell proliferation and death, on cell fate and tumor growth.

## 2. Materials and Methods

### 2.1. Human HCCs

Surgically obtained liver tumor samples of 67 patients with HCC were analyzed. The specimens were obtained under patient written informed consent, following a procedure that was approved by the local ethical committee. Samples were snap frozen and maintained at −80 °C until use. Demographic and clinical data of the patients are reported in Appendix A. After surgery, patients were regularly followed up, and clinical, laboratory and imaging assessment were recorded, as previously described [27].

### 2.2. Cell Lines

HepG2 cell line was authenticated by BMR Genomics S.r.l. (Padova, Italy), according to PowerPlex^®^ Fusion System protocol (Promega) and was regularly tested for mycoplasma contamination. This cell line was stably transfected with the full-length human SERPINB3 genomic sequence or with the plasmid vector alone, which were used as previously described [16]. The experiments were carried out using transfected clone 2 (HepG2/SB3) and were confirmed using transfected HepG2 clone 3 [16]. An additional HepG2 clone, stably transfected with a Reactive Site Loop (RSL) deleted-SerpinB3 plasmid (Δ-SerpinB3) (provided by Dr. Tim J. Harrison, UCL Medical School, London), was also used to assess the functional role of the antiprotease activity of this serpin in our experimental conditions. Cells were maintained at 37 °C in a humidified chamber with 5% CO_2_ and cultured in minimum essential medium with the addition of G418 as selective agent.

### 2.3. Quantitative Real-Time RT–PCR

Total RNA was extracted using RNasy Trizol (Invitrogen, Carlsbad, CA, USA) according to the manufacturer’s instructions. After determination of the purity and the integrity, total RNA, complementary DNA synthesis and quantitative real-time PCR reactions (RT-PCR) were carried out as previously described [19] using the CFX96 real-time instrument (Bio-Rad Laboratories Inc, Hercules, CA, USA). In hepatoma cells and HCC samples the relative expression was generated for each sample by calculating 2-Δ Ct [28]. Primers sequences used in the study are reported in Appendix A.

### 2.4. Western Blot Analysis

Total protein contents (50 μg) from each cellular extract, prepared at 4 °C in lysis buffer (150 mM NaCl, 10 mM Tris-HCl (pH 7.4), 1 mM EDTA, 1 mM EGTA, 2% Triton X-100) in the presence of phosphatase and protease inhibitors (Roche, IN, USA), were loaded onto 10% polyacrylamide gel. The blots were probed with the following primary antibodies: rabbit polyclonal anti-p66shc (Upstate Cell Signaling Solution, NY, USA), rabbit oligoclonal anti-SerpinB3 (Hepa Ab, Xeptagen, Venice, Italy), mouse anti-p66 shc (Upstate Cell Signaling Solution, NY, USA), rabbit polyclonal anti-β-Catenin (GeneTex, Irvine, CA, USA), rabbit polyclonal anti-LC3B (Abcam, Cambridge, UK), rabbit polyclonal anti-active-Caspase-8 (Novus Biologicals-Bio-Techne, Centennial, CO, USA), mouse monoclonal anti-RIP3K (Santa Cruz biotechnology, Dallas, TX, USA). Mouse monoclonal anti-β actin (Sigma-Aldrich, St. Louis, MO, USA) was used as housekeeping control. Anti-mouse and anti-rabbit horseradish peroxides-conjugated antibodies (Amersham, Arlington Heights, IL, USA) were used as secondary antibodies. Antigenic detection was carried out by enhanced chemiluminescence (Amersham, Arlington Heights, IL, USA) and densitometric analysis was assessed using the VersaDoc Imaging System (Bio-Rad Laboratories, Hercules, CA, USA). Relative density units were obtained by comparing ratios of intensities (volume as sum of the intensities of the pixels inside the volume boundary x area of a single pixel (in mm^2^)) of a reference band (β-Actin).

### 2.5. Immunohistochemistry

Quantification of macrophage infiltration was assessed by immunohistochemistry on paraffin-embedded tumor xenografts using anti-F4/80 monoclonal antibody (Abcam). Results were analyzed in 5 slides per group, counting positive cells in 10 fields per slide. The slides were observed blindly by two operators and then by a third operator using a bright field microscope.

### 2.6. Immunofluorescence

The expression of SerpinB3 and of p66shc in SerpinB3 transfected cells and in controls was assessed by immunofluorescence. Cells were seeded on slides (2 × 10^5^ HepG2 cells/slide) and cultured for 48 h. After fixation with 4% paraformaldehyde and permeabilization with 0.2% Tryton X100, cells were blocked with 5% BSA in PBS. The slides were incubated with anti-SerpinB3 antibody (8 μg/mL), or with anti-p66 Shc antibody (1:50 dilution), washed with 0.1% Tween 20 in phosphate-buffered saline (PBS) and incubated with Alexa Fluor 488 Goat anti-mouse and Alexa Fluor 546 Goat anti-rabbit (Invitrogen, Thermo Fisher, Waltham, MA, USA) as secondary antibodies.

Cellular nuclei were counter-stained with Hoechst 33,342 (Sigma-Aldrich, St. Louis, MO, USA). The slides were mounted with ELVANOL (Sigma-Aldrich, St. Louis, MO, USA) and observed under a fluorescence microscope (Axiovert 200M, Carl Zeiss MicroImaging GmbH, Gottingen, Germany).

### 2.7. Apoptosis/Necrosis Detection Assay

An Apoptosis/Necrosis detection kit (Abcam) was used to simultaneously monitor apoptotic, necrotic and healthy cells through staining with Apopxin Green Solution (green Ex/Em = 490/525 nm) to detect phosphatidylserine (PS) as marker of initial/intermediate stages of apoptosis, with 7-AAD (/-aminoactinomycin D) (red Ex/Em = 546/647 nm) to detect loss of plasma integrity, characteristic of late apoptosis and necrosis. Briefly, 8 × 10^5^ HepG2/SB3 and HepG2/Ctrl cells were seeded onto coverslip and grown until semi-confluence. After overnight treatment with 100 μM H_2_O_2_, cells were washed twice and incubated with staining probes and incubated at room temperature for 1 h. After washing, the slides were mounted with ELVANOL (Sigma-Aldrich, St. Louis, MO, USA) and observed under a fluorescence microscope (Axiovert 200M, Carl Zeiss MicroImaging GmbH, Gottingen, Germany).

### 2.8. Cell Viability Assay

The effect of oxidative stress on cell viability was estimated using the MTT assay. Cells were cultured in 96-well plates (40,000 cells/well) and treated with H_2_O_2_ as specified, followed by incubation at 37 °C for 5 h to determine cell viability. In the MTT assay, after dissolving formazan crystals, light absorbance was measured at 570 nm using a spectrophotometer (Victor, Perkin Elmer, Waltham, MA, USA). The quantity of formazan product in the culture medium was directly proportional to the number of viable cells.

### 2.9. Flow Cytometric Analysis of Cell Death

Cells seeded at 2 × 10^5^/well were treated with different concentrations of diamide as indicated and incubated for 2.5 or 18 h. After treatment, cells were washed with phosphate-buffered saline (PBS) and resuspended in 50 µL HEPES buffer (10 mM HEPES, 135 mM NaCl, 5 mM CaCl_2_). Cells were then incubated for 15 min at 37 °C with Propidium Iodide (PI, 1 µg/mL). Samples were analyzed on a FACSCalibur flow cytometer (Becton Dickinson, Franklin Lakes, NJ, USA). Data acquisition was performed using CellQuest software (Becton Dickinson, Franklin Lakes, NJ, USA) and data analysis was carried out with WinMD free software.

### 2.10. Cell Proliferation

The xCELLigence system (Roche Diagnostics GmbH, Mannheim, Germany and ACEA Biosciences, Inc., San Diego, CA, USA) was used to evaluate cell proliferation, according to the instructions of the supplier. A cell suspension of 5 × 10^4^ cells was seeded in each well of E-plate 16 in quadruple, cultured in complete MEM and maintained in a CO_2_ incubator at 37 °C with 95% humidity and 5.0% carbon dioxide saturation. After 18 h, cells were treated with H_2_O_2_ diluted in MEM or with MEM alone as control, and automatically monitored every 5 min for 48 h. Dynamic cell proliferation of cells plated was monitored at 5 min intervals from the time of plating until the end of the experiment, analyzed with RTCA software and expressed as cell index value.

### 2.11. Measurement of Intracellular Ca^2+^

HepG2 cells overexpressing SerpinB3 and control HepG2 were seeded onto coverslips (3 × 1 cm) and allowed to grow to confluence. The medium was then changed to medium without serum and the cells were used after 24 h. Before starting the experiments, the cells were loaded with 3 µmol/L Fura-2 AM for 1 h at room temperature. Then, Fura-2 was removed, a physiological medium (containing in mmol/L: NaCl 129, KCl 2.8, KH_2_PO_4_ 0.8, CaCl_2_ 1, NaHCO_3_ 8.9, MgCl_2_ 0.8, glucose 5.6, HEPES 5.6; pH 7.4) was added, and cells were incubated at room temperature for 30 min. The coverslip was placed into a quartz cuvette (3 mL) inside a fluorescent spectrophotometer (Shimadzu-1501) equipped with a thermostatted cuvette holder and superfused with physiological medium at 37 °C. The baseline fluorescence was obtained by rapidly alternating the excitation wavelength between 340 and 380 nm and recording the 510 nm emission intensity. Ca^2+^ levels were calculated according to the standard formula: (Ca^2+^) = Kd (R-Rmin)/(Rmax-R)/(Sf2/Sb2). Kd was taken as 224 nmol/L, and Rmax, Rmin and Sf2/Sb2 were calculated by plotting a calibration curve with buffers containing various Ca^2+^ concentrations.

### 2.12. Calcium Retention Capacity (CRC)

The CRC assay has been used to assess the propensity to open of the mitochondrial permeability transition pore (mtPTP), as previously described [29]. Briefly, HepG2/SB3 and HepG2/Ctrl cells, after diamide treatment, were permeabilized with 100 μM digitonin (15 min, 4 °C) in a 1 mM EGTA buffer. Digitonin was then eliminated and permeabilized cells were placed in 10 μM EGTA in the presence of the Ca^2+^ probe Calcium Green-5N (1 µM; λ exc: 505 nm; λ em: 535 nm; Molecular Probes), which does not permeate mitochondria. Cells were then exposed to Ca^2+^ spikes (5 μM), and fluorescence drops were used to assess mitochondrial Ca^2+^ uptake using a Fluoroskan Ascent FL (Thermo Electron Corp. Thermo Fisher, Waltham, MA USA) plate reader. PTP opening was detected as a sudden and irreversible fluorescence increase due to Ca^2+^ release from mitochondria.

### 2.13. Murine Experimental Models

*p66Shc Knockout Mice*. The p66Shc knockout mice, corresponding to the ShcP strain (provided by M. Giorgio, Department of Biomedical Sciences, University of Padova, Padova, Italy), a recognized model of long survival [4], and the corresponding wild-type C57BL/6 mice strain were bred at the animal facility of the Venetian Institute of Molecular Medicine (Padua, Italy). The liver of 16 females, 18–24-week-old mice was used to investigate the p66shc extent of expression in relation to different molecules involved in cell death and survival, including SerpinB3, β-catenin and LC3b.

*Xenograft Models*. To assess the effect of SerpinB3 overexpression on tumor growth, 26 Scid mice (carrying a genetic immune deficiency that affects B and T cells) and 12 Rag-c57 mice (depleted of NK cells) were injected subcutaneously with HepG2/SB3 cells or HepG2/control cells, previously transfected with luciferase, at 1 × 10^6^ concentration (right- and left-hand flanks). Tumor growth was monitored weekly using a BioLuminescent Imaging instrument. Mice were sacrificed at day 25after injection, tumors were removed and morphological and molecular analyses were carried out.

Expression of p66shc, SerpinB3, pro-inflammatory (IL-6 and TNF-α) and anti-inflammatory (IL-10) cytokines were analyzed by real-time PCR. The macrophage marker F4/80 was evaluated by immunohistochemistry. Protein expression of p66shc, SerpinB3, β-Catenin, activated Caspase-8, whole and cleaved RIP3K were investigated by Western blot.

All the experiments, conducted in accordance with the “Principles of laboratory animal care” (NIH publication no. 85–23, revised 1985; http://grants1.nih.gov/grants/olaw/references/phspol.htm (accessed on 10 January 2011)), were approved by the local ethical committee and the Italian Ministry of Health.

### 2.14. Statistical Analysis

Statistical analysis was performed by Student’s *t*-test or ANOVA for analysis of variance when appropriate (*p* < 0.05 was considered significant). Spearman’s rank correlation was used to measure statistical dependence between two variables. For molecular variables a cut-off value ≥ median value was defined as “high-mRNA expression” to be considered as a dichotomous categorical variable, while cases with values < median value, were defined as “low-mRNA expression”. The overall time of survival curves was calculated using the Kaplan–Meier method and compared using the log-rank test. Data in bar graphs represent means ± SEM and were obtained from at least three independent experiments. Western-blot and morphological images are representative of at least three experiments with similar results.

## 3. Results

### 3.1. SerpinB3 Increases the Negative Influence of p66shc on Survival of Patients with Hepatocellular Carcinoma

Overall, the mRNA expression of p66shc in liver tumors was inversely correlated with the expression of SerpinB3 (Figure 1A). The pattern of expression of these two molecules was differently distributed within patients with HCC (Figure 1B), although patients with low p66shc levels showed longer survival than those with high expression (Figure 1C). It is interesting to note that while for patients with low levels of SerpinB3, survival was not affected by p66shc (Figure 1E), patients with low levels of p66shc and high levels of SerpinB3 showed a markedly better survival (Figure 1D), while the presence of high levels of both molecules was associated with significantly shorter survival (Figure 1D). No significant modification of overall survival was observed in relation to SerpinB3 expression alone, while a trend of better survival was detected in patients with high SerpinB3 and low p66shc (Appendix A). These results suggest that SerpinB3 can influence p66shc-related fate, making these two molecules strictly linked with clinical outcome. The worst survival rate, observed when both p66shc and SerpinB3 were high, was associated with the concomitant presence of high levels of β-catenin and low expression of TGF-β1 (Figure 2A,D), two molecules involved in HCC prognosis [20], but only lower β-catenin levels were associated with better survival (Figure 2B,D).

### 3.2. SerpinB3 and p66shc Are Inversely Correlated in Hepatoma Cell Lines and in p66shc-/- Mice

In HepG2 cells stably transfected with the full-length human SerpinB3 genomic sequence, p66shc was significantly reduced both at the transcription and protein levels (Figure 3). These results were confirmed also in an additional SerpinB3-transfected HepG2 cell clone (HepG2/SB3 clone 3, Appendix A). The inhibitory effect of SerpinB3 on p66shc expression was also observed in cells transfected to overexpress SerpinB3 lacking the reactive loop (Δ-SerpinB3) but expressing the same amounts of SerpinB3 at transcription and protein level (Figure 3), indicating that the antiprotease activity of SerpinB3 is dispensable for the modulation of p66shc expression in these experimental conditions.

An inverse correlation between p66shc and SerpinB3 was also detected in p66shc knock-out mice (Figure 3A), since they presented increased levels of SerpinB3, while this molecule was barely detectable in control mice (Figure 3C). As reported in Figure 4, the *p66Shc*-/- mice showed also a decrease of the autophagy marker LC3B and of the LC3BI/LC3BII ratio, as previously described [11], associated with lower β-catenin expression.

### 3.3. SerpinB3 and p66shc Affect Cell Susceptibility to Death in Different Oxidative Stress Conditions

In order to examine the effect of SerpinB3 and p66shc on cell death/survival in different oxidative stress conditions, increasing concentrations of a physiological oxidant (hydrogen peroxide) and of a chemical oxidizing agent (diamide) were used. As expected, SerpinB3-expressing cells exhibited higher proliferation in basal conditions and higher survival at low (50 μM) H_2_O_2_ (Appendix A). However, at higher H_2_O_2_ concentrations, a remarkable drop in cell viability was observed in HepG2/SB3, which was significantly higher than that observed in control HepG2 cells (Figure 5A). In keeping with these results, a more pronounced decrease of normalized cell index was observed in HepG2/SB3 cells at high H_2_O_2_ concentration (Figure 5B). Furthermore, by simultaneously monitoring apoptosis and necrosis through cell labelling with fluorescent probes, a larger number of apoptotic/necrotic cells was found in HepG2/SB3 than in wild-type HepG2 cells (Figure 5C).

Furthermore, diamide treatment determined a higher incidence of cell death in HepG2/SB3 than in HepG2 wild-type cells (Figure 6A,B).

Ca^2+^ is a key effector of intracellular signaling, and a dysregulated elevation of cytoplasmic Ca^2+^ can lead to cell death. A significantly higher concentration of cytoplasmic Ca^2+^ was observed in cells overexpressing SerpinB3 (Figure 6C), which suggests that these cells are more sensitive to cell death stimuli. Many such stimuli converge on inducing the opening of a channel called the mitochondrial permeability transition pore (mtPTP), a point of no return in the cell death process [29]. In agreement with our previous findings, a greater propensity of mtPTP to open after diamide treatment was observed in HepG2/SB3 cells (Figure 6D).

### 3.4. Low p66shc and High SerpinB3 Decrease the Growth of Tumor Xenografts in Mice

Subcutaneous tumor xenografts were generated in immunodeficient mice to verify the effect of the pattern characterized by low p66shc and high SerpinB3 in tumor growth. Scid mice inoculated with HepG2/SerpinB3 cells showed significantly lower tumor growth, with weight and volume of tumor masses at sacrifice 5 and 6 times lower, respectively, than those detected in mice injected with HepG2/control cells (Figure 7A–C).

Similar growth profiles were also observed in Rag-c57 mice (N. 4 for each group) that were injected with HepG2/SB3 clone 2 or with HepG2/SB3 clone 3 or with HepG2 transfected with the plasmid vector alone (Appendix A), ruling out a possible involvement of the NK compartment of the immune system in these findings. Transcripts and protein expression from tumor tissue of mice inoculated with HepG2 cells overexpressing SerpinB3 not only confirmed the overexpression of SerpinB3 and the lower values of p66shc but also showed barely detectable levels of β-catenin, at variance with controls, where its expression was significantly higher (Figure 8A–C and Appendix A).

This molecular profile was reminiscent of that observed in liver tumors of patients showing the best survival (Figure 1). To investigate the mode of cell death, activated Caspase-8 was analyzed, since apoptosis and necroptosis may follow TNF-α signaling depending on whether Caspase-8 is active or inactive, respectively [25]. Western blot showed a marked decrease of active Caspase-8 in tumors obtained in mice injected with HepG2/SB3 (Figure 8D and Appendix A), suggesting a possible activation of necroptosis. This occurrence was confirmed by the finding of a concomitant upregulation of RIP3K protein and decreased levels of cleaved RIP3K, a result of Caspase-8 inactivation (Figure 8E,F and Appendix A). In keeping with these findings, tumors originating from HepG2/SB3-injected cells showed also an overall increased expression of the pro-inflammatory cytokines IL-6 and TNF-α, compared to tumors originating from HepG2/Control cells, while the levels of the anti-inflammatory cytokine IL-10 remained unchanged (Figure 8G). This inflammatory profile was associated with decreased macrophage infiltration (Figure 8H,I), which is typical of necroptosis [26].

## 4. Discussion

In this study we have analyzed the relationship between p66shc and SerpinB3 expression in hepatocellular carcinoma in relation to clinical outcome and its potential involvement in cell death. Both molecules have been shown to control cell death with opposite effects, although p66shc, known for its pro-apoptotic function due to ROS production [4,5], has been recently identified as an important factor implicated in the regulation of the autophagic process, which may rather lead to lymphocyte survival [12]. In the absence of p66shc the autophagic flux is impaired and lower levels of LC3II are observed, as also detected in our *p66shc−/−* mice, where SerpinB3 was found to be overexpressed. SerpinB3 is an anti-apoptotic molecule commonly detected in tumors with worse survival, including HCC [20,21,22].

In our cohort of patients with HCC, p66shc has been found to be associated with the clinical outcome. The worst overall survival was observed when p66shc was highly expressed, as already shown in previous studies [12]. Matching different levels of expression of p66shc and SerpinB3, we identified two subgroups of patients characterized by a clearly distinct behavior in terms of survival rate. The worst survival rate was observed when both p66shc and SerpinB3 were highly expressed and this pattern was associated with the concomitant presence of high levels of β-catenin, notoriously involved in malignant transformation, often in parallel with SerpinB3 [20,22,30]. This subgroup was also characterized by low levels of TGF-β1 that were similar to those observed in the subgroup of patients with the best prognosis where, however, the high levels of SerpinB3 were associated with low p66shc expression. These findings are in keeping with the fact that β-catenin, but not TGF-β1, was differently expressed in relation to patients’ survival.

The sharp difference in clinical outcome observed in surgically resected patients with high levels of SerpinB3, but different extent of p66shc expression in tumoral tissue, has potentially interesting implications for management of patients with HCC, since different therapeutic approaches could be proposed based on the expression pattern of the two molecules. For example, liver transplantation could be considered especially for those patients with high levels of both SerpinB3 and p66shc.

On the basis of these premises, we have investigated how the p66shc/SerpinB3 relationship could affect cell death. In cultured cell lines, as also observed in *p66shc−/−* mice, an inverse relationship between the presence of SerpinB3 and p66shc was documented. In HepG2 cells constitutively expressing p66shc, this molecule actually was markedly reduced when enforced SerpinB3 expression was induced. When cells overexpressing SerpinB3 and low levels of p66shc were exposed to different extents of oxidative stress conditions, an opposite behavior was observed in relation to ROS concentration. Indeed, they were more resistant to apoptosis at low H_2_O_2_ concentration, as previously reported [16], but, surprisingly, these cells were more prone to die compared to control cells expressing low SerpinB3 and high p66shc levels, in the presence of a high concentration of pro-oxidant stimuli such as H_2_O_2_ and diamide. Higher Ca^2+^ concentrations were detected, which may act as a trigger of the mitochondrial permeability transition pore (mtPTP), resulting in earlier cell death [29]. In line with this prediction, when subcutaneously injected into the xenograft model, cells overexpressing SerpinB3 and low p66shc induced smaller tumors that also showed trivial levels of β-catenin, supporting the better prognosis observed in the subgroup of patients with a similar pattern to these two molecules. In addition, these tumors were characterized by inhibition of Caspase-8 activity that, in a proinflammatory background, could make cells more sensitive to necroptosis, a recently described TNF-α induced form of programmed cell death [31]. The high levels of TNF-α, the accumulation of receptor-interacting protein kinase-3 (RIP3K) and its lack of cleavage by Caspase-8 in mice injected with HepG2/SB3 provide evidence of necroptotic death in this experimental model, highlighting its protective function in slowing down tumor growth.

This scenario suggests a novel beneficial function of SerpinB3 through induction of cell death by necroptosis when p66shc is downregulated in conditions of oxidative stress. Further studies are required to better understand the microenvironment and cellular factors responsible for the different expression profiles of these molecules found in tumor specimens. Cancer cells often acquire the ability to evade cell death induced by chemotherapeutic agents, and induction of necroptosis could represent an efficient way to overcome cell-death resistance [32].

## 5. Conclusions

In conclusion, our findings identified a new molecular pattern in patients affected by HCC where elevated levels of SerpinB3 associated with p66shc downregulation showed markedly better survival, likely because tumor cells were more prone to die by necroptosis. These findings further highlight the importance of markers, allowing a tailored approach to patients with cancer, not only to personalize therapy but also to define the optimal management, improving the clinical outcome.

## Figures and Tables

**Figure 1 biology-10-00363-f001:**
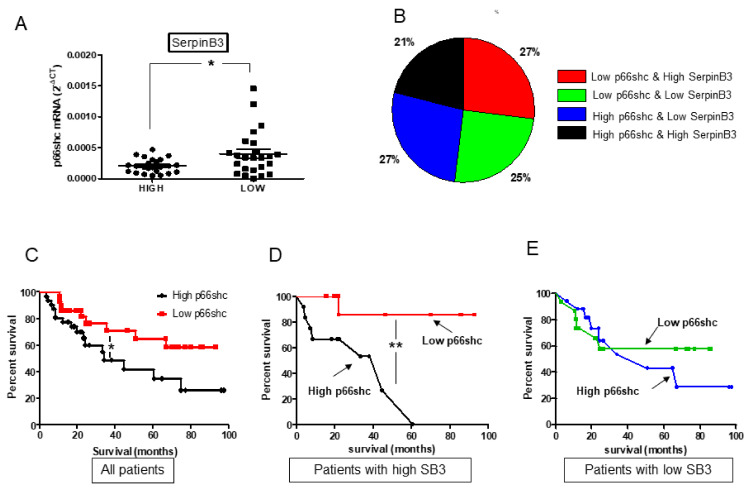
P66shc and SerpinB3 expression in relation to survival in patients with hepatocellular carcinoma. (**A**) Relative p66shc mRNA levels detected in tumor samples of patients with “High SerpinB3” or “Low SerpinB3” (* *p* = 0.016); (**B**) Distribution of the different patterns of p66shc and SerpinB3 in HCC specimens in relation to the high or low expression of these two molecules defined as “high” when values were > of the median value or “low” when the values were < of the median value. (**C**) Kaplan–Meier survival curves in the cohort of 67 patients with HCC, divided into two groups according to high or low p66shc mRNA expression level (* *p* = 0.0465); (**D**) Kaplan–Meier survival curves of two subgroups of HCC patients with high expression of SerpinB3 (SB3) and high or low expression of p66shc (** *p* = 0.039); (**E**) Kaplan–Meier survival curves in two subgroups of HHC patients with low expression of SerpinB3 (SB3) and high or low expression of p66shc (*p* = NS).

**Figure 2 biology-10-00363-f002:**
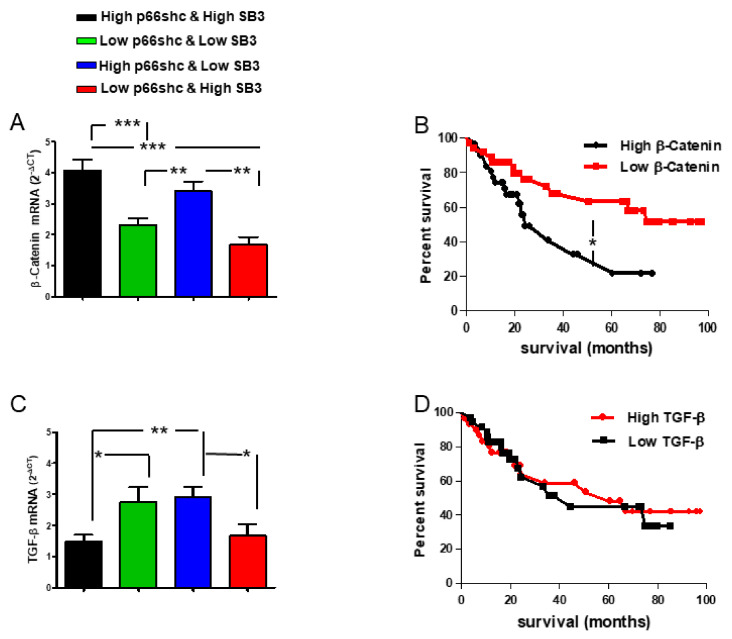
β-catenin and TGF-β1 expression in relation with p66shc and SerpinB3. (**A**) Relative β-catenin mRNA expression levels in four HCC subgroups divided on the basis of different expression levels of p66shc and SerpinB3. β-catenin was significantly higher in HCC patients with high p66shc, regardless of SerpinB3 expression (“High p66shc and High SB3” vs. “Low p66shc and low SB3” *** *p* = 0.0002, vs. “Low p66shc and High SB3” *** *p* < 0.0001; “High p66shc and Low SB3” vs. “Low p66shc and Low SB3” ** *p* = 0.0091, vs. “Low p66shc and High SB3” ** *p* = 0.0014). (**B**) Kaplan–Meier survival curves in HCC patients divided into two groups on the basis of high or low β-Catenin mRNA expression levels (* *p* = 0.0224). (**C**) Relative TGF-β expression levels in the four HCC subgroups divided on the basis of different expression levels of p66shc and SerpinB3. TGF-β1 expression was significantly lower in HCC patients with High SerpinB3, regardless of p66shc expression (“High p66shc and high SB3” vs. “Low p66shc and low SB3” * *p* = 0.0372, vs. “High p66shc and low SB3” ** *p* = 0.009; “Low p66shc and High SB3 vs. “High p66shc and Low SB3” * *p* = 0.0185, “High p66shc and High SB3” vs. “Low p66shc and High SB3” *p* = NS); (**D**) Kaplan–Meier survival curves in HCC patients divided into two groups on the basis of high or low TGF- β1 mRNA expression levels (*p* = NS).

**Figure 3 biology-10-00363-f003:**
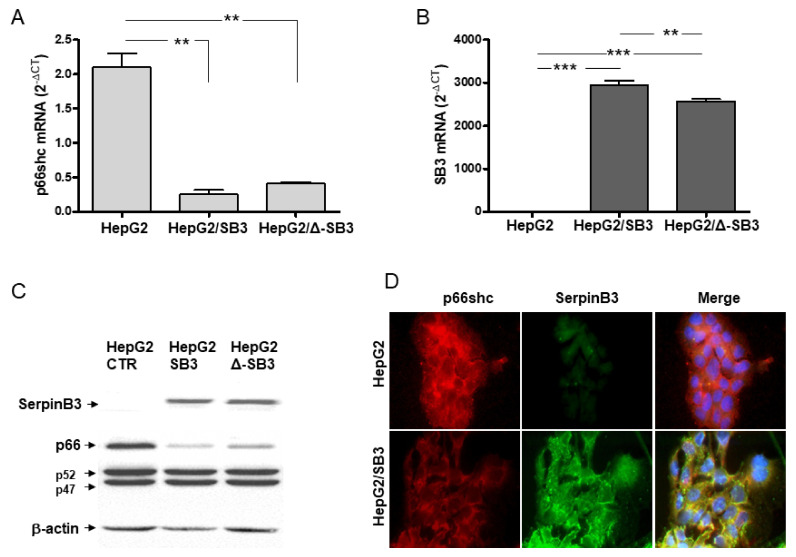
Inverse correlation of p66shc and SerpinB3 expression in hepatoma cells. Graphs representing mRNA levels, expressed as 2^-ΔCT^, for p66shc (**A**) and for SerpinB3 (**B**) in control HepG2 cells (HepG2), in HepG2 transfected to overexpress wild-type SerpinB3 (HepG2/SB3) or Reactive Site Loop deleted-SerpinB3 (HepG2/Δ-SB3); examples of SerpinB3 and of p66shc protein expression in control HepG2 and in both wild-type HepG2/SB3 or HepG2/Δ-SB3, detected by Western blot (**C**) or by immunofluorescence (**D**) are reported. ** *p* < 0.01; *** *p* < 0.001.

**Figure 4 biology-10-00363-f004:**
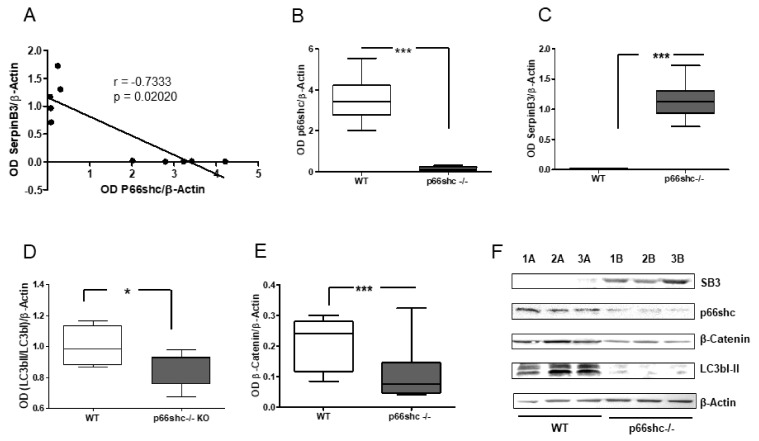
Inverse correlation of p66shc and SerpinB3 expression in *p66shc*-/- KO mice. (**A**) Inverse correlation between SerpinB3 and p66shc protein expression in mice (wild type and KO for *p66shc*); graphs representing densitometric analysis of Western blots for p66shc (**B**), SerpinB3 (**C**), LC3b I/LC3bI ratio (**D**) and β-catenin (**E**), protein expression normalized to GAPDH in liver specimens of *p66shc*-/- KO mice and in the relative wild-type (WT) controls; (**F**) Representative example of Western blot of these proteins in three wild-type mice (1A–3A) and in three p66shc K mice (1B–3B). * *p* < 0.05; *** *p* < 0.001.

**Figure 5 biology-10-00363-f005:**
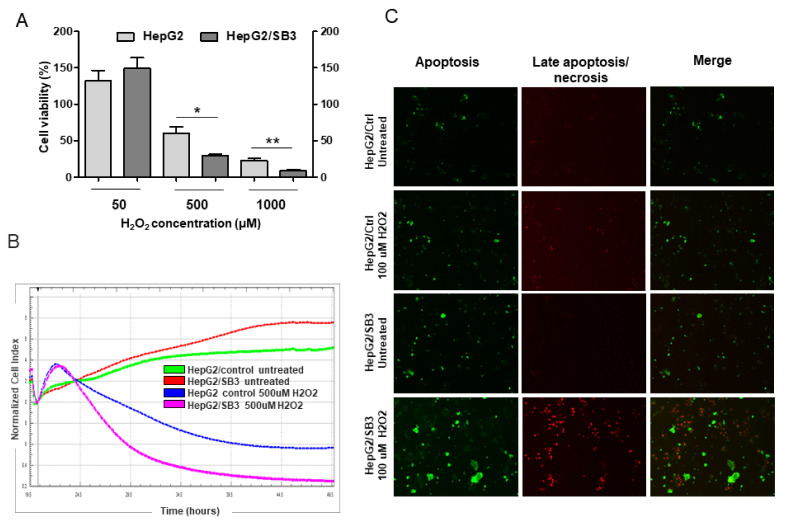
Effect of H_2_O_2_ at different concentrations on cell proliferation and viability in HepG2 cells with different SerpinB3 and p66shc expression. (**A**) Cell viability assay: HepG2/SB3 and control HepG2 cells were exposed to increasing concentrations of H_2_O_2_ for 5 h and cell viability was assessed by MTT assay. Data represent the mean + SD of three independent experiments. (**B**) Cell proliferation curve: The xCELLigence system was used to evaluate cell proliferation. After seeding 100 μL (5 × 10^4^ cells/well) of the cell suspensions into the 16-well E-plate, untreated cells or cells treated with H_2_O_2_ diluted in MEM at different concentrations (50 uM, 1 mM) were monitored every 5 min for 18 h. After 5 h of 1 mM H_2_O_2_ treatment, cells overexpressing SerpinB3 (HepG2/SB3) showed a significant decrease in viability, compared to controls. (**C**) Fluorescence analysis for cell death: HepG2/SB3 cells and control HepG2/Ctrl cells after overnight treatment with 100 uM H_2_O_2_ and stained with fluorescent probes for apoptosis (green, Apopxin Green Indicator) and late apoptosis/necrosis (red, 7-AAD) are shown. A greater presence of cells with signs of both apoptosis and late apoptosis/necrosis after treatment is detectable in HepG2/SB3 cells, compared to controls. * *p* < 0.05; ** *p* < 0.01.

**Figure 6 biology-10-00363-f006:**
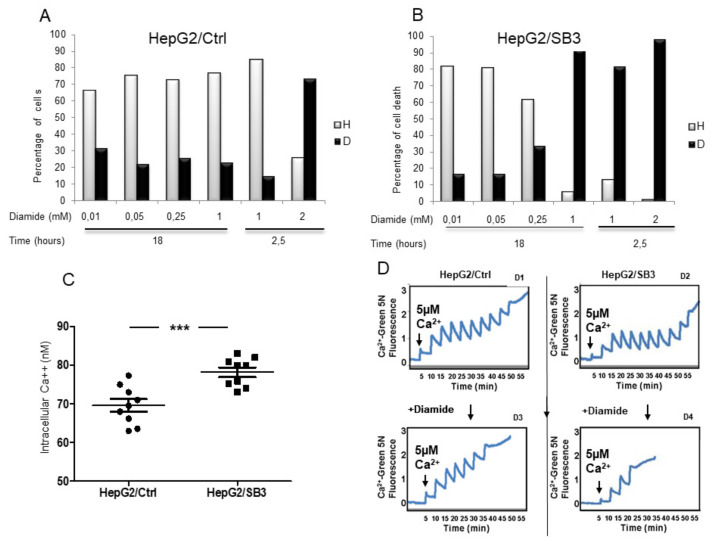
Effect of diamide at different concentrations on cell death in HepG2 cells with different SerpinB3 and p66shc expression. (**A**) Percentage of dead cells (D, black) or healthy cells (H, gray) detected by FACS analysis after treatment with increasing concentrations of diamide in HepG2/CTR cells with low levels of SerpinB3 and high levels of p66shc (**A**) and in cells overexpressing SerpinB3 but lacking p66shc (HepG2/SB3) (**B**); (**C**) Extent of intracellular calcium concentration in different HepG2/SB3 and HepG2/Ctrl cells (*** *p* < 0.001); (**D**) Calcium retention capacity after Ca^2+^ spikes in absence or presence of diamide, affecting PTP opening in HepG2/SB3 and HepG2/Ctrl cells. After diamide treatment, lower calcium concentrations are needed to determine PTP opening in HepG2/SB3 (D4), compared to controls (D3).

**Figure 7 biology-10-00363-f007:**
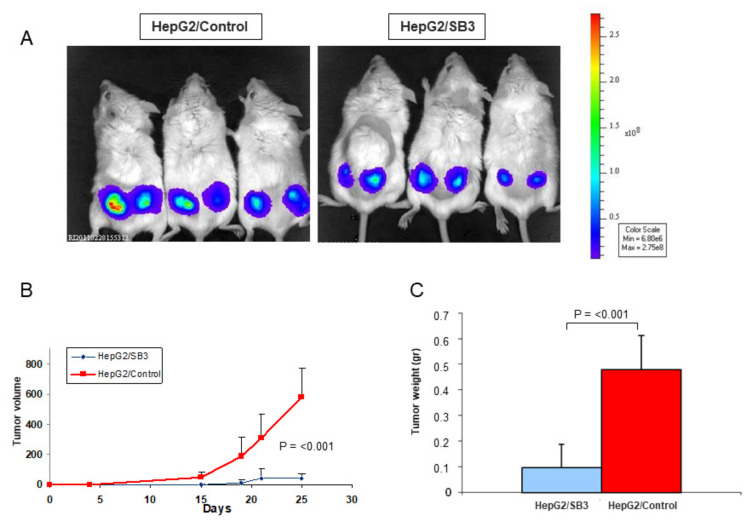
Tumor growth in Scid mice injected with HepG2 cells expressing different extent of SerpinB3 and p66shc. (**A**) Bioluminescence detected in representative Scid mice inoculated with luciferase-treated HepG2/control cells (3/13 mice) or HepG2/SB3 cells (3/13 mice) are shown, on the right the color scale is reported; (**B**) Overtime tumor growth detected during 25 days of follow-up; each point represents median tumor volume of all 13 mice per group, bars represent standard error; (**C**) Median tumor weight detected at sacrifice in the two experimental groups of 13 mice each. Bars represent standard error.

**Figure 8 biology-10-00363-f008:**
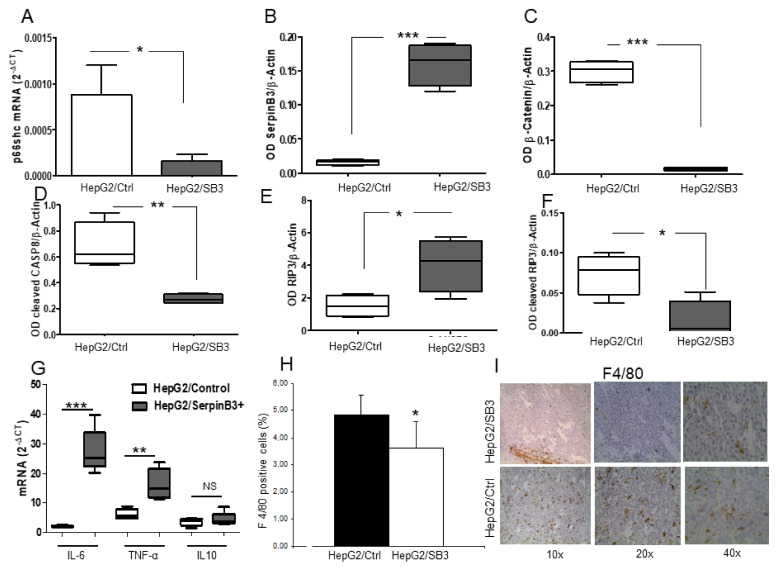
Cell death markers in tumor xenografts. Expression of p66shc (**A**), SerpinB3 (**B**), β-Catenin (**C**), active caspase-8 (**D**), whole (**E**) and cleaved RIP3K (**F**) levels in tumors of Scid mice inoculated with HepG2/SB3 cells or with control HepG2 cells (HepG2/Ctrl); (**G**) Inflammatory cytokines profile of total tumor mass in tumors derived from HepG2/SB3 (dark gray) and in those derived from HepG2/Ctrl cells (white), ** *p* <0.01, *** *p* < 0.001; (**H**) Extent of macrophage infiltration after F4/80 immunostaining in tumors originated from HepG2/control and from HepG2/SB3-injected cells (* *p* < 0.05, t student). Columns represent mean values and bars standard error; (**I**) Representative images of F4/80 immunostaining in section of tumor specimen of mice injected with HepG2/SB3 and in mice injected with HepG2/Ctrl cells. The image shows reduced infiltration of macrophages in HepG2/SB3 compared with the HepG2/Ctrl tumor. Three different magnifications (10×, 20×, 40×) are shown.

## Data Availability

The data presented in this study are available on request from the corresponding author. The data are not publicly available due to privacy reasons.

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
