# Peer review of "Low P66shc with High SerpinB3 Levels Favors Necroptosis and Better Survival in Hepatocellular Carcinoma"

_biology, 2021, doi:10.3390/biology10050363_

Round 1

Reviewer 1 Report

Comment

The manuscript entitled “Low P66shc With High Serpinb3 Levels Favours Necroptosis And Better Survival In Hepatocellular Carcinoma” described a comprehensive molecular, cellular, in vivo mouse modeling approach with a reflection on the clinical aspect. It is of great interest.

The manuscript, however, did not provide complete sets of data, specifically HepG2 cells with different SerpinB3 and p66shc expression (Fig 7). Simultaneously, only SerpinB3 was used for molecular analyses (Fig 8), which did not support the title. Such an asymmetric design, especially with a limited number of mice ( N = 3), down-graded the study's significance.

Some locations need a transition point to articulate where the problem ends, and the solution begins - to make sure that the manuscript flows from one part to the next. Below specific comments and suggestions for the authors should be incorporated to improve its clarity, coherence, and logic flow.

Specific comments:

  • Lines 22-23, Abstract: “An inverse expression of p66shc and SerpinB3 was detected both in p66shc-/- mice and in HepG2 cells overexpressing SerpinB3.” It is of confusion in the logic. Explain the relevant quantification of so-called low p66shc and high SerpinB3 levels in the human vs. mouse model - “In patients with HCC the best survival was observed in the subgroup with low p66shc and high SerpinB3 levels.” (Lines 21-22). Define an internal control for “low p66shc and high SerpinB3 levels”.
  • Line 25: “necroptosis” – “the pro-apoptotic p66shc and the anti-apoptotic SerpinB3” – Note that tumor necrosis and necroptosis differ from apoptosis. Any biomarkers did they look at for typical features of necroptosis relative to other cell death modes?
  • Lines 23 – 24: “These cells were more prone to die under conditions of stress and developed smaller tumors in nude mice.” The authors need to be specific for “these cells” or “What stress” or “smaller tumors” – how much smaller compared with what models?
  • L56-58: “SerpinB3 has been found to be overexpressed in several types of tumors, especially in those with poor prognosis, including breast, liver, esophagus, and colorectal cancer [19-22], although some findings have suggested its ability to inhibit cancer cell invasion [23].” Not sufficient introduction to explain the bridge to their title: “Low P66shc With High Serpinb3 Levels Favours Necroptosis And Better Survival In Hepatocellular Carcinoma.”
  • L60-65: Not sufficient introduction to explain the difference between Necroptosis (necrosis) and apoptosis. E.g., necrosis and apoptosis represent two pathways of genetically encoded necrotic cell death and apoptotic cell death. “Necroptosis is a non-caspase-dependent and precisely regulated mechanism of cell death. Necroptosis serves as an alternative mode of programmed cell death overcoming apoptosis resistance and may trigger and amplify antitumor immunity in cancer therapy.”
  • L72: “liver tumor samples of 67 patients with HCC were analyzed” – a table for demographics data should be provided, including the duration of tumors.
  • Fig 1A: “A) Relative p66shc mRNA levels detected in tumor samples of patients with “High SerpinB3” or “Low SerpinB3” (*p = 0.016)” – what was the unit of the scale (0.0000 to 0.0005 to 0.0010)? The mean value is about 0.000025 between high and low” – how did they draw the line? How did they define “High SerpinB3” or “Low SerpinB3?” Where was the data? Fig 1C, D, E: X-axis unit, days? Months?
  • Line 243: “For patients with low levels of SerpinB3, survival was not affected by p66shc (Figure 1E).” Data for levels of SerpinB3? How could they jump to “These results suggest that SerpinB3 can influence p66shc-related fate, making these two molecules strictly linked with clinical outcome.” (L244-245).
  • Fig2A, “β-catenin was significantly higher in HCC patients with high p66shc, regardless of SerpinB3 expression (“High p66shc & High SB3” vs “Low p66shc & low SB3” p = 0.0002, vs “Low p66shc & High SB3” p =<0.0001.” L247-248: “but only β-catenin levels impacted significantly on survival (Figure 2 B, D)” – How did they define “impacted?” It is more accurate to state “associated with” as they did not establish the caustic effects rather than correlated results.
  • L342-343: “Figure 7. Tumor growth [in] Scid mice injected with HepG2 cells expressing the different extent of SerpinB3 and p66shc.”
  • L344: “HepG2/control cells (three mice) or HepG2/SB3 cells (three animals)” – Only 3 mice, what’s the power of the statistics? They need to provide the individual values of the 3 mice.
  • L348-349: “with the plasmid vector alone (Suppl. Figure 3)” “Rag-c57 mice” – How many mice? Why with different values?
  • Fig 8I, the scale bars are needed for all the panels with descriptions of the morphology. Fig8B, C, D, E, F should come with Western blotting to validate a specific protein's identity.
  • Lines 367-368: “Western blot showed a marked decrease of active Caspase-8 in tumors obtained in mice injected with HepG2/SB3 (Figure 8D)” – Where was the original WB?
  • Lines 358-359: “G) Inflammatory cytokines expressions in tumors derived from HepG2/SB3 (dark gray) and in those derived from HepG2/Ctrl cells (white).” Tumors comparisons? It is not logical to compare the heterogeneous tissues without confirming the homogenous cells – such as cell compositions based on single-cell analyses. E.g., F4/80 immunostaining in tumors (Fig 8H) shows the error bars are overlapped.
  • L384-385: “SerpinB3 is an anti-apoptotic molecule commonly described as a negative prognostic factor in different tumor types, including HCC [20, 21,22].” Define what they wanted to say using “negative prognostic factor?” They further stated (L391-394): “The worst survival rate was observed when both p66shc and SerpinB3 were highly expressed and this pattern was associated with the concomitant presence of high levels of β-catenin, notoriously involved in malignant transformation, often in parallel with SerpinB3 [20,22,29].”

Author Response

POINT BY POINT REPLY TO THE REVIEWER 1

We are grateful for the overall comments of the reviewer that expressed a great interest for the manuscript and provided us the opportunity to improve several points of the paper. In detail are reported the reply to specific comments:

Lines 22-23, Abstract: “An inverse expression of p66shc and SerpinB3 was detected both in p66shc-/- mice and in HepG2 cells overexpressing SerpinB3.” It is of confusion in the logic.

These sentence has been changed with “Mice p66shc-/- showed high levels of SerpinB3, while in HepG2 cells overexpressing SerpinB3, p66shc expression was trivial. HepG2 overexpressing SerpinB3 cells were more prone to die after oxidizing treatments, such as diamide or high concentration H2O2”.  

Lines 21-22Explain the relevant quantification of so-called low p66shc and high SerpinB3 levels in the human vs. mouse model.

The definition of low and high levels of SerpinB3 in humans was changed in “median values below or above median values”.

Line 25: Note that tumor necrosis and necroptosis differ from apoptosis. Any biomarkers did they look at for typical features of necroptosis relative to other cell death models?

As biomarkers of necroptosis the sentence “decreased activated Caspase-8, with concomitant increase of RIP3K and decreased levels of cleaved RIP3K” was added.

Lines 23 – 24: These cells were more prone to die under conditions of stress and developed smaller tumors in nude mice.” The authors need to be specific for “these cells” or “What stress” or “smaller tumors” – how much smaller compared with what models?

Details have been added to the sentence: “HepG2 overexpressing SerpinB3 cells were more prone to die after oxidizing treatments, such as diamide or high concentration H2O2. These cells injected in nude mice developed tumors 5 times smaller than those from control HepG2 cells treatments.

Lines 56-58: “SerpinB3 has been found to be overexpressed in several types of tumors, especially in those with poor prognosis, including breast, liver, esophagus, and colorectal cancer [19-22], although some findings have suggested its ability to inhibit cancer cell invasion [23].” Not sufficient introduction to explain the bridge to their title: “Low P66shc With High Serpinb3 Levels Favours Necroptosis And Better Survival In Hepatocellular Carcinoma.”

The following sentence “These finding suggest that additional players might modulate the biological activity of this molecule, influencing cell fate.” has been added to explain the bridge to the title.

L60-65: Not sufficient introduction to explain the difference between Necroptosis (necrosis) and apoptosis.

The introduction has been extended explaining the difference between Necroptosis (necrosis) and apoptosis, following the Reviewer suggestion and we are very grateful for this improved  definition.

L72: “liver tumor samples of 67 patients with HCC were analyzed” – a table for demographics data should be provided, including the duration of tumors.

As requested, the median survival time of the patients has been added in Supplemental Table 1.

Fig 1A: “A) Relative p66shc mRNA levels detected in tumor samples of patients with “High SerpinB3” or “Low SerpinB3” (*p = 0.016)” – what was the unit of the scale?... how did they draw the line?.. How did they define “High SerpinB3” or “Low SerpinB3?” Where was the data? Fig 1C, D, E: X-axis unit, days? Months?

The unit scale has been added  (2-DCT) and the description has been added in the legend of the figure. As defined in the Statistical analysis a cut-off value > median value was defined as “high-mRNA expression”, while cases with values < median value, were defined as “low-mRNA expression”, now this information has been added in the legend of the Figure 1. Raw data of quantitative PCR of p66sch mRNA have been provided  as additional  file  for the reviewer.                                                               In the axis unit “months” has been added.

Line 243: “For patients with low levels of SerpinB3, survival was not affected by p66shc (Figure 1E).” Data for levels of SerpinB3?

Kaplan -Meier survival curves in relation to SerpinB3 levels have been provided in Suppl. Figure 1 and the sentence “No significant modification of overall survival was observed in relation to SerpinB3 expression alone, while a trend to better survival was detected in patients with high SerpinB3 and low p66shc” has been added to the first part of the results.

Lanes 244-245: How could they jump to “These results suggest that SerpinB3 can influence p66shc-related fate, making these two molecules strictly linked with clinical outcome.”

The sentence has been extended to better support the conclusion “while for patients with low levels of SerpinB3, survival was not affected by p66shc, patients with low levels of p66shc and high levels of SerpinB3 showed a markedly better survival”.

L247-248: “but only β-catenin levels impacted significantly on survival (Figure 2 B, D)” – How did they define “impacted?” It is more accurate to state “associated with”..

We acknowledge the suggestion of the reviewer and the sentence has been modified with “only lower β-catenin levels were associated with better survival”.

L342-343: “ .. Tumor growth [in] Scid mice injected with HepG2 cells expressing the different extent of SerpinB3 and p66shc.”

The title of the paragraph has been modified according with the reviewer suggestion.

L344: “HepG2/control cells (three mice) or HepG2/SB3 cells (three animals)” – Only 3 mice, what’s the power of the statistics? They need to provide the individual values of the 3 mice.

In the Figure 7A only 3 out 13 mice per group have been included as representative images and this information has been added to the legend of Figure 7. In figure 7B and 7C the results of median tumor volume and weight are referred to all the 13 mice included in each group, as now speficified in the legend of the figure.

L348-349: “with the plasmid vector alone (Suppl. Figure 3)” “Rag-c57 mice” – How many mice? Why with different values?

The suppl.Figure (now suppl. Figure 4) represents the median tumor volume (bars represent SE)  in Rag-c57 mice (N. 4 for each group) that were injected with HepG2/SB3 clone 2, or 3  (used as additional clone overexpressing SerpinB3) and  with the vector alone, as reference control. 

Fig 8I, the scale bars are needed for all the panels with descriptions of the morphology.

The description of the morphology of Figure 8I has been now better detailed in the legend of Figure 8I and also the specification that three different magnifications (10x, 20x, 40x) are shown, has been added.

Fig 8 B, C, D, E, F should come with Western blotting to validate a specific protein's identity.

Western blotting of all the panels have been provided in supplemental Figure 5 and original blots are available for the reviewers.

Lines 358-359: “G) Inflammatory cytokines expressions in tumors derived from HepG2/SB3 (dark gray) and in those derived from HepG2/Ctrl cells (white).” Tumors comparisons? It is not logical to compare the heterogeneous tissues without confirming the homogenous cells – such as cell compositions based on single-cell analyses. . E.g., F4/80 immunostaining in tumors (Fig 8H) shows the error bars are overlapped.

The results of Figure 8G are expressed as” total tumor mass expression” of different cytokines in tumors derived from HepG2 cells overexpressing or not SerpinB3, it would be interesting having also cell compositions based on single-cell analyses, but unfortunately we were not able to get this information at the time when the experiments were run.  We acknowledge that the bars were overlapped and now the figure 8H has been corrected.

L384-385: “SerpinB3 is an anti-apoptotic molecule commonly described as a negative prognostic factor in different tumor types, including HCC [20, 21,22].” Define what they wanted to say using “negative prognostic factor? The sentence in the They further stated (L391-394): “The worst survival rate was observed when both p66shc and SerpinB3 were highly expressed and this pattern was associated with the concomitant presence of high levels of β-catenin, notoriously involved in malignant transformation, often in parallel with SerpinB3 [20,22,29].”

Discussion has been now changed with “in tumors with worse survival” to better understand the message. The sentence of Lanes 391-394 is now better supported by the above change, suggested by the reviewer.

POINT BY POINT REPLY TO THE REVIEWERS

Reviewer 1

We are grateful for the overall comments of the reviewer that expressed a great interest for the manuscript and provided us the opportunity to improve several points of the paper. In detail are reported the reply to specific comments:

Lines 22-23, Abstract: “An inverse expression of p66shc and SerpinB3 was detected both in p66shc-/- mice and in HepG2 cells overexpressing SerpinB3.” It is of confusion in the logic.

These sentence has been changed with “Mice p66shc-/- showed high levels of SerpinB3, while in HepG2 cells overexpressing SerpinB3, p66shc expression was trivial. HepG2 overexpressing SerpinB3 cells were more prone to die after oxidizing treatments, such as diamide or high concentration H2O2”.  

Lines 21-22Explain the relevant quantification of so-called low p66shc and high SerpinB3 levels in the human vs. mouse model.

The definition of low and high levels of SerpinB3 in humans was changed in “median values below or above median values”.

Line 25: Note that tumor necrosis and necroptosis differ from apoptosis. Any biomarkers did they look at for typical features of necroptosis relative to other cell death models?

As biomarkers of necroptosis the sentence “decreased activated Caspase-8, with concomitant increase of RIP3K and decreased levels of cleaved RIP3K” was added.

Lines 23 – 24: These cells were more prone to die under conditions of stress and developed smaller tumors in nude mice.” The authors need to be specific for “these cells” or “What stress” or “smaller tumors” – how much smaller compared with what models?

Details have been added to the sentence: “HepG2 overexpressing SerpinB3 cells were more prone to die after oxidizing treatments, such as diamide or high concentration H2O2. These cells injected in nude mice developed tumors 5 times smaller than those from control HepG2 cells treatments.

Lines 56-58: “SerpinB3 has been found to be overexpressed in several types of tumors, especially in those with poor prognosis, including breast, liver, esophagus, and colorectal cancer [19-22], although some findings have suggested its ability to inhibit cancer cell invasion [23].” Not sufficient introduction to explain the bridge to their title: “Low P66shc With High Serpinb3 Levels Favours Necroptosis And Better Survival In Hepatocellular Carcinoma.”

The following sentence “These finding suggest that additional players might modulate the biological activity of this molecule, influencing cell fate.” has been added to explain the bridge to the title.

L60-65: Not sufficient introduction to explain the difference between Necroptosis (necrosis) and apoptosis.

The introduction has been extended explaining the difference between Necroptosis (necrosis) and apoptosis, following the Reviewer suggestion and we are very grateful for this improved  definition.

L72: “liver tumor samples of 67 patients with HCC were analyzed” – a table for demographics data should be provided, including the duration of tumors.

As requested, the median survival time of the patients has been added in Supplemental Table 1.

Fig 1A: “A) Relative p66shc mRNA levels detected in tumor samples of patients with “High SerpinB3” or “Low SerpinB3” (*p = 0.016)” – what was the unit of the scale?... how did they draw the line?.. How did they define “High SerpinB3” or “Low SerpinB3?” Where was the data? Fig 1C, D, E: X-axis unit, days? Months?

The unit scale has been added  (2-DCT) and the description has been added in the legend of the figure. As defined in the Statistical analysis a cut-off value > median value was defined as “high-mRNA expression”, while cases with values < median value, were defined as “low-mRNA expression”, now this information has been added in the legend of the Figure 1. Raw data of quantitative PCR of p66sch mRNA have been provided  as additional  file  for the reviewer.                                                               In the axis unit “months” has been added.

Line 243: “For patients with low levels of SerpinB3, survival was not affected by p66shc (Figure 1E).” Data for levels of SerpinB3?

Kaplan -Meier survival curves in relation to SerpinB3 levels have been provided in Suppl. Figure 1 and the sentence “No significant modification of overall survival was observed in relation to SerpinB3 expression alone, while a trend to better survival was detected in patients with high SerpinB3 and low p66shc” has been added to the first part of the results.

Lanes 244-245: How could they jump to “These results suggest that SerpinB3 can influence p66shc-related fate, making these two molecules strictly linked with clinical outcome.”

The sentence has been extended to better support the conclusion “while for patients with low levels of SerpinB3, survival was not affected by p66shc, patients with low levels of p66shc and high levels of SerpinB3 showed a markedly better survival”.

L247-248: “but only β-catenin levels impacted significantly on survival (Figure 2 B, D)” – How did they define “impacted?” It is more accurate to state “associated with”..

We acknowledge the suggestion of the reviewer and the sentence has been modified with “only lower β-catenin levels were associated with better survival”.

L342-343: “ .. Tumor growth [in] Scid mice injected with HepG2 cells expressing the different extent of SerpinB3 and p66shc.”

The title of the paragraph has been modified according with the reviewer suggestion.

L344: “HepG2/control cells (three mice) or HepG2/SB3 cells (three animals)” – Only 3 mice, what’s the power of the statistics? They need to provide the individual values of the 3 mice.

In the Figure 7A only 3 out 13 mice per group have been included as representative images and this information has been added to the legend of Figure 7. In figure 7B and 7C the results of median tumor volume and weight are referred to all the 13 mice included in each group, as now speficified in the legend of the figure.

L348-349: “with the plasmid vector alone (Suppl. Figure 3)” “Rag-c57 mice” – How many mice? Why with different values?

The suppl.Figure (now suppl. Figure 4) represents the median tumor volume (bars represent SE)  in Rag-c57 mice (N. 4 for each group) that were injected with HepG2/SB3 clone 2, or 3  (used as additional clone overexpressing SerpinB3) and  with the vector alone, as reference control. 

Fig 8I, the scale bars are needed for all the panels with descriptions of the morphology.

The description of the morphology of Figure 8I has been now better detailed in the legend of Figure 8I and also the specification that three different magnifications (10x, 20x, 40x) are shown, has been added.

Fig 8 B, C, D, E, F should come with Western blotting to validate a specific protein's identity.

Western blotting of all the panels have been provided in supplemental Figure 5 and original blots are available for the reviewers.

Lines 358-359: “G) Inflammatory cytokines expressions in tumors derived from HepG2/SB3 (dark gray) and in those derived from HepG2/Ctrl cells (white).” Tumors comparisons? It is not logical to compare the heterogeneous tissues without confirming the homogenous cells – such as cell compositions based on single-cell analyses. . E.g., F4/80 immunostaining in tumors (Fig 8H) shows the error bars are overlapped.

The results of Figure 8G are expressed as” total tumor mass expression” of different cytokines in tumors derived from HepG2 cells overexpressing or not SerpinB3, it would be interesting having also cell compositions based on single-cell analyses, but unfortunately we were not able to get this information at the time when the experiments were run.  We acknowledge that the bars were overlapped and now the figure 8H has been corrected.

L384-385: “SerpinB3 is an anti-apoptotic molecule commonly described as a negative prognostic factor in different tumor types, including HCC [20, 21,22].” Define what they wanted to say using “negative prognostic factor? The sentence in the They further stated (L391-394): “The worst survival rate was observed when both p66shc and SerpinB3 were highly expressed and this pattern was associated with the concomitant presence of high levels of β-catenin, notoriously involved in malignant transformation, often in parallel with SerpinB3 [20,22,29].”

Discussion has been now changed with “in tumors with worse survival” to better understand the message. The sentence of Lanes 391-394 is now better supported by the above change, suggested by the reviewer.

Reviewer 2 Report

Summary:

The authors present an interesting article titled “Low P66shc With High Serpinb3 Levels Favours Necroptosis And Better Survival In Hepatocellular Carcinoma”. It demonstrates a correlation in patients regarding P66shc and Serpinb3. p66shc null mice experience an enhanced serpinb3 RNA and protein. Further, there is a physiological effect to cells with high Serpinb3, where cells experience enhanced cell death. However, the authors must make several major and minor revisions before acceptance.

Major revisions:

  1. For Figure 1, please add data to demonstrate the survival curves to replicate Figure 1C for high and low serpin3.
  2. For Figure 1, please add data to demonstrate the survival curves to replicate Figure 1D and E, but specifically looking at “Patients with high p66shc-1D” and “Patients with low p66shc-1D”.
  3. For Figure 3, Δ-SerpinB3 overexpression experiments need to demonstrate via western blot as well.
  4. Figure 4F westerns need to be redone for B-catenin, p66shc, and serpinb3. The quality is not adequate for publication.
  5. For the p66shc, a rescue experiment would benefit the manuscript to demonstrate that serpinb3 levels are directly regulated by p66shc.
  6. The authors need to knockdown or knockout Serpinb3 in a cell line that expresses high serpinb3, to identify the endogenous role of this protein. Does p66shc get affected with loss of endogenous sb3?
  7. Overall, the authors must go over all figure legends and double check to make sure their writing matches the figure images. There are several errors noted in the minor revisions, and I may have missed some during my reading.

Minor revisions:

  1. Line 25, please change to “…necroptosis. In conclusion,…”.
  2. Please ensure the text font and size is consistent throughout entire manuscript.
  3. Line 247, please change in the manuscript “Figure 2A,C”.
  4. Line 340, change in the manuscript “Figure 7A,B,C”.

Author Response

POINT BY POINT REPLY TO  REVIEWER 2

 We are grateful to the reviewer for his interest in the manuscript and for his suggestions that have allowed a remarkable contribution to the improvement of the paper. The following points have been addressed:

  1. Data to demonstrate the survival curves to replicate Figure 1C for high and low SerpinB3 have been provided in Supplemental Figure 1 A and the sentence “No significant modification of overall survival was observed in relation to SerpinB3 expression alone, while a trend to better survival was detected in patients with high SerpinB3 and low p66shc” has been added to the first part of the results.
  2. Data to demonstrate the survival curves to replicate Figure 1D and E, but specifically looking at “Patients with high p66shc-1D” and “Patients with low p66shc-1D have been added in Supplemental Figure 1B and 1C.
  3. For Figure 3, Western blot including also the effect of Δ-SerpinB3 has now been added to Figure 3C. Original blot has also been provided for review.
  4. The available data are those obtained in the liver of p66shc -/- mice, showing that the lack of p66shc is associated with detectable levels of SerpinB3, not detectable in the corresponding wild type strain of mice.
  5. Additional rescue experiments would be advisable to demonstrate that serpinb3 levels are directly regulated by p66shc, as suggested by the reviewer, but unfortunately, they are not available at the present time.
  6. The knockdown or knockout SerpinB3 experiments could indeed reinforce the present findings, however, the limited time for reviewing did not allow to carry on this type of experiments.
  7. All figure legends have been carefully double checked and the mistakes have been corrected.

The minor revisions suggestions have been addressed.

Round 2

Reviewer 1 Report

accepted.

Reviewer 2 Report

No additional comments.